



# Reviews and syntheses: Greenhouse gas emissions from drained organic forest soils – synthesizing data for site-specific emission factors for boreal and cool temperate regions

Authors: Jyrki Jauhiainen[1], Juha Heikkinen[1], Nicholas Clarke[2], Hongxing He[3], Lise Dalsgaard[2], Kari Minkkinen[4], Paavo Ojanen[1,4], Lars Vesterdal[5], Jukka Alm[1], Aldis Butlers[6], Ingeborg Callesen[5], Sabine Jordan[7], Annalea Lohila[8,9], Ülo Mander[10], Hlynur Óskarsson[11], Bjarni D. Sigurdsson[11], Gunnhild Søgaard[12], Kaido Soosaar[10], Åsa Kasimir[13], Brynhildur Bjarnadottir[14], Andis Lazdins[6], and Raija Laiho[1]

[1] Natural Resources Institute Finland (Luke), Box 2, FI–00791 Helsinki, Finland

[2] Department of Terrestrial Ecology, Norwegian Institute of Bioeconomy Research (NIBIO), Box 115, N–1431 Ås, Norway

[3] Department of Geography, McGill University, QC H3A 0B9 Montréal, Canada

[4] Department of Forest Sciences, University of Helsinki, Box 27, FI–00014, Helsinki, Finland

[5] Department of Geosciences and Natural Resource Management, University of Copenhagen, DK–1958 Frederiksberg C, Denmark

[6] Latvian State Forest Research Institute (Silava), Salaspils, LV–2169, Latvia

[7] Department of Soil and Environment, Swedish University of Agricultural Sciences, P.O. Box 7014, 75007 Uppsala

[8] INAR Institute for Atmospheric and Earth System Research/Physics, Faculty of Science, University of Helsinki, Box 68, FI-00014, Helsinki, Finland

[9] Finnish Meteorological Institute, Climate System Research, Box 503, FI–00101 Helsinki, Finland

[10] Department of Geography, University of Tartu, EE-51014 Tartu, Estonia

[11] Agricultural University of Iceland, IS-311 Hvanneyri, Borgarnes, Iceland

[12] Department of Forest and Climate, Norwegian Institute of Bioeconomy Research (NIBIO), Box 115, N–1431 Ås, Norway

[13] Department of Earth Sciences, University of Gothenburg, Box 460, SE–40530 Gothenburg, Sweden

[14] Department of Education, University of Akureyri, IS-600 Akureyri, Iceland

*Correspondence to:* Jyrki Jauhiainen (jyrki.jauhiainen@luke.fi)

*Keywords:* peatlands, organic soils, carbon dioxide, methane, nitrous oxide, peat, gleysol, muck, peatland forests, forestry drainage, afforestation

**Abstract.** We compiled published peer-reviewed $CO_2$, $CH_4$ and $N_2O$ data on drained organic forest soils in boreal and temperate zones, to revisit the current Tier 1 default emission factors (EFs) provided in the IPCC (2014) Wetlands Supplement: to see whether their uncertainty may be reduced, to evaluate possibilities for breaking the broad categories used for the IPCC EFs into more site-type specific ones, and to inspect the potential relevance of a number of environmental variables for predicting the annual soil greenhouse gas

(GHG) balances, on which the EFs are based. Despite a considerable number of publications applicable for compiling EFs was added, only modest changes were found compared to the Tier 1 default EFs. However, the more specific site-type categories generated in this study showed narrower confidence intervals compared to



the default categories. Overall, the highest $CO_2$ EFs were found for temperate afforested agricultural lands, and boreal forestry-drained sites with very low tree-stand productivity. The highest $CH_4$ EFs in turn prevailed at boreal nutrient-poor forests with very low tree-stand productivity and temperate forests irrespective of nutrient status, while the EFs for afforested sites were low or showed a sink function. The highest $N_2O$ EFs were found for afforested agricultural lands and forestry-drained nutrient-rich sites. The occasional wide confidence intervals could be mainly explained by single or few highly deviating estimates, rather than the broadness of the categories applied. Our EFs for the novel categories were further supported by the statistical models connecting the annual soil GHG balances to site-specific soil nutrient status indicators, tree stand characteristics, and temperature-associated weather and climate variables. The results of this synthesis have important implications for EF revisions and national emission reporting, e.g., by the use of different categories for afforested sites and forestry-drained sites, and more specific site productivity categories based on timber production potential.

## 1 Introduction

Organic soils, characterized by large deposits of organic carbon (C) and nitrogen (N), have potentially large fluxes of the greenhouse gases (GHG) carbon dioxide ($CO_2$), methane ($CH_4$) and nitrous oxide ($N_2O$). Wetlands characterised by organic soils (we hereafter use 'wetlands') are found in all climate zones, but most distributed in the Nordic (Denmark, Finland, Iceland, Norway, Sweden) and Baltic (Estonia, Latvia, Lithuania) countries in boreal and cool temperate climate zones (Montanarella et al., 2006; Köchy et al., 2015; Conchedda and Tubiello, 2020). Peat is the most common organic soil formed in wetlands classified as peatlands (histosols), but also other organic soil types (often identified as gleysols, gyttja or muck) can be found. Globally, peatlands have been widely subjected to land-use change for agriculture and forestry or peat extraction (Joosten, 2010). Of the peatlands and other wetlands in the EU, c.a. 20 % are under anthropogenic land use and land-use change (UNFCCC, 2017).

Wetlands, typically characterized by a high soil water-table level (i.e., distance of the water table in soil from soil surface, from here onwards we use WT), are usually drained to improve aeration and nutrient availability for crops grown in agriculture and trees in forestry, or to ease peat extraction. Lowered WT enhances aerobic decomposition in organic matter and thus the mobilization of C and N stores in organic soils (e.g., Post et al., 1985; Kasimir-Klemedtsson et al., 1997; Ernfors et al., 2008; Petrescu et al., 2015; Abdalla et al., 2016; Pärn et al., 2018). Drainage and land-cover changes together alter rates in several processes: biomass growth, dead organic matter (litter) inputs into the soil, and litter and soil organic matter decomposition. Measurements on changes in soil and vegetation C stocks and GHG flux rates over time can be used for estimating the ecosystem GHG balance.

Drained organic soils are identified as a significant source of atmospheric GHG emissions in the national inventories under international treaties (IPCC, 2014; Oertel et al., 2016; Wilson et al., 2016). Currently, both the IPCC (2006) agriculture, forestry, and other land use (AFOLU) guidance and the IPCC (2014) Wetlands Supplement may be used for reporting annual GHG emissions and removals for soils under anthropogenic land uses, including forests growing on drained organic soils. Area-based emission factors (EFs), describing the net annual GHG ($CO_2$, $CH_4$ and $N_2O$) emissions or removals, have been developed for different land management and environmental conditions. IPCC (2014) provided default EFs for drained organic forest soils (Table 1). The $CO_2$ EF for forest soils describes the annual difference between the amount of C added into the soil as above- and belowground litter, and the $CO_2$ efflux resulting from the decomposition of litter and soil, the estimation depending on the monitoring method used in data collection (Jauhiainen et al., 2019). EFs for $CH_4$ and $N_2O$ are based on the net gas exchange between the soil surface and the atmosphere.





Countries may opt for different methodological levels in their GHG reporting by applying the default IPCC EFs (Tier 1), EFs based on country-specific data (Tier 2) or repeated national inventories and/or advanced modelling (Tier 3). The Tier 1 EFs for drained organic forest soils are average emission values based on peer-reviewed studies covering a wide range of situations categorized by climatic zones, and at the most detailed
level the EFs are specified for nutrient-poor versus nutrient-rich conditions (Table 1). However, such simple classification lumps together a wide range of conditions and forest types differing in vegetation communities, C-input rates and GHG efflux from decomposition processes. For the temperate zone, there is only one EF based on an average of all published emission rates.

The high uncertainties in the Tier 1 EFs motivate to develop Tier 2 and Tier 3 EFs and to use existing data
more efficiently (IPCC, 2014; Oertel et al., 2016; Tubiello et al., 2016; Kasimir et al., 2018). The C balance in the soil is linked to soil nutrient status as the vegetation that contributes to the C balance through tree stand growth, species composition, and thus the amount and quality of litter deposited on the forest floor and below ground differ between nutrient-rich and nutrient-poor sites (e.g. Minkkinen et al., 1999; Ojanen et al., 2010, 2013, 2019; Uri et al., 2017). Environmental factors influencing the annual release of $CO_2$ from decomposition
include temperature conditions that differ from south to north with faster decomposition in the south (Ojanen et al., 2010), and oxygen availability for aerobic decomposers (Pearce and Clymo, 2001; Jaatinen et al., 2008; Ojanen and Minkkinen, 2019). Increasing depth of the oxic soil layer in the growing season, controlled by (artificial) drainage, increases the biomass of aerobic decomposers, fungi and bacteria, in peat soils (Jaatinen et al., 2008) especially on the nutrient-rich sites (Peltoniemi et al., 2021). To further develop EFs, the essential
environmental variables controlling emissions must be identified, monitored, and reported regularly (Jauhiainen et al., 2019).

Drained organic forest soils generally vary from a small source to a small sink of $CH_4$ (e.g., Ojanen et al., 2010; Rütting et al., 2014). Methane, formed by methanogenic archaea in anoxic, water-saturated soil layers, forms an energy source for methanotrophic bacteria in oxic soil layers, the extent of which depend on WT
(Sundh et al., 1994; Larmola et al., 2010), and the proportion of oxidized $CH_4$ is avoided emission. Further, plant species composition is linked to soil $CH_4$ balance, and, especially, the abundance of sedges is a good predictor of $CH_4$ emissions (Nykänen et al., 1998; Dias et al., 2010; Couwenberg and Fritz, 2012; Turetsky et al., 2014). The plant communities in different site types and drainage conditions can thus result in different levels and direction of $CH_4$ transfer between soil and the atmosphere.

High soil N content and redox conditions that vary between oxidative and reductive are favourable for $N_2O$ production in organic soils (Regina, 1998; Pärn et al., 2017). Drainage-induced decomposition creates favourable conditions for the nitrification of ammonium ($NH_4^+$) to nitrate ($NO_3^-$). This process leads to $N_2O$ production due to inefficient processing to ammonium. If there is an excess of $NO_3^-$ (not by vegetation) under wet conditions, $N_2O$ may be produced from $NO_3^-$ denitrification back to $NH_4^+$ in such reductive conditions.
Thus, N-rich drained sites with temporal changes in water saturation have the highest potential for $N_2O$ emissions (Martikainen et al., 1993; Ojanen et al., 2013; Minkkinen et al., 2020).

Most drained organic forest soils are a result of the drainage of natural peatlands for forestry, i.e., they are "forestry-drained" sites. However, some drained organic forest soils have been formed by afforesting former agricultural lands (i.e., pastures and croplands) or cutaway peat extraction areas by seeding or planting trees.
Afforested agricultural lands have likely been subject to frequent tilling, fertilisation and other soil amendments (e.g., sand or lime) for up to several decades, which have modified the soil nutrient status. Afforested peat extraction areas normally have only the millennia-old bottom peat left, possibly mixed with some underlying mineral matter. Management history may thus possess legacy effects, which potentially change the soil GHG exchange levels in comparison to forestry-drained sites (e.g., Meyer et al., 2013;
Mäkiranta et al., 2007; Lohila et al., 2007). Further, different types of organic soils (e.g., peat, muck, gyttja)



may have different soil GHG exchange levels due to differences in physical or chemical soil properties. The most common soil type may vary regionally, e.g., andosols are more common than histosols in Iceland (Arnalds, 2015). So far, many GHG studies have been carried out on organic soils that were not explicitly classified as peat (e.g., Ball et al., 2007; Mander et al., 2008; McNamara et al., 2008; Christiansen et al., 2012), but Tier 1 EFs are an average of data from all organic soil types together (IPCC 2014). Uncertainty in Tier 1 EFs can thus be expected to decrease both by an increase in the number of soil GHG balance estimates in the present categories, and by using more specific categories.

Part of the uncertainty in the IPCC $CO_2$ EFs for drained organic forest soils may derive from differences between GHG flux monitoring methods (chamber- vs. eddy covariance techniques) and soil C stock inventory methods (see e.g., Jauhiainen et al., 2019). In IPCC (2014) flux and stock inventory data were pooled together. Gaseous flux monitoring enables linking the monitored $CO_2$ dynamics to the environmental conditions prevailing during the monitoring period, whereas the estimate in soil C stock inventories is a net change in soil C-stocks over decades-long periods in the past. The soil $CO_2$ balance estimates based on these fundamentally different methods could, and should, be compared if sufficient data representing comparable site conditions exist.

Soil chemical and physical characteristics, vegetation community (tree stand composition and stock, ground vegetation), weather and climate (e.g., amount and distribution of precipitation and temperature), and position in the landscape (e.g., altitude, latitude) may influence soil C and N dynamics and are to a varying extent included in the publications collected for this study (Jauhiainen et al., 2019). However, to our knowledge no previous study has systematically assessed the correlations of soil GHG balance estimates with these characteristics. If significant correlations can be found, they may be utilized in modelling and developing dynamic Tier 3 EFs.

We compiled published peer-reviewed soil $CO_2$, $CH_4$ and $N_2O$ exchange data for drained organic forest soils in the boreal and temperate zones to evaluate to what extent these data would allow the development of higher-tier EFs. We focused on data that have been used, or have the potential to be used, for estimating annual soil GHG balances as in the IPCC (2014) Wetlands Supplement. From here onwards we use 'annual soil GHG balance' for site-level estimates, and 'emission factor (EF)' for estimates pooled into different site-type categories as averages. Our goals were to investigate: (a) how the EFs and the EF uncertainties differ between site-type specific categories and the broad categories applied in the IPCC Tier 1 EFs, i.e., whether this uncertainty can be reduced by the use of more specific categories, (b) comparability of $CO_2$ EFs based on flux data and inventory data, (c) potential sources of EF uncertainty detectable in the data, and (d) to which extent the site-specific annual soil GHG balances correlate with site-specific variables for weather, climate, soil and vegetation, and thus could serve the development of models aiming at higher EF tiers.

## 2 Materials and methods

We compiled $CO_2$, $CH_4$ and $N_2O$ flux data from peer-reviewed literature focusing on drained organic forest soils in boreal and temperate cool and moist climate area as defined in IPCC (2006). We utilized the data base compiled by Jauhiainen et al. (2019) and complemented it until the end of the year 2019. The methods applied in the $CO_2$ data collection in the assessed publications included repeated soil C stock inventories and GHG flux measurements by chambers and by eddy covariance, and thus 'inventory data' and 'flux data' were identified in the database (S1). All $CH_4$ and $N_2O$ estimates were based on flux data collected by closed chambers. For chamber methods, each annual soil $CO_2$ balance estimate for a site is the estimated net outcome of C fluxes into the soil in above-ground and below-ground litter, and C losses in the decomposition of litter and soil organic matter, for a one-year period. For eddy covariance studies, the annual soil $CO_2$ balance is





estimated as net ecosystem exchange minus net primary production. For soil inventory methods, annualized
soil C stock change is used as the estimate. 'Annual soil $CH_4$ and $N_2O$ balance' estimates are annual
cumulative GHG fluxes based on flux data collection.

## 2.1 Criteria for data selection

Organic soils were defined as thickness of the surface soil organic layer of at least 10 cm and a minimum
organic C content of 12% by mass, even if the soil is mixed to a depth of 20 cm (as in Annex 3A.5, IPCC
2006). In practice these organic soils included peat and soils identified as gleysols or muck (histosol with
sapric soil material). The organic soils other than peat were collectively named 'other organic soils'. For the
definition of forests, we included sites specified as forests in the original publications unless the described site
characteristics differed from the specifications applied in IPCC (2014), where the minimum criteria are forest
canopy coverage of at least 10 % of the area, continuous forest area size more than 0.5 ha, and trees with a
minimum height of 5 m in maturity on the site (as in FRA 2015).

The studied forests were assumed to be under typical management, so sites with excessive experimental
fertilization or extreme hydrology intervention were excluded. If a site was afforested after another land use,
a period of at least 20 years as forest land was used as a criterion (as in IPCC 2014). Forests on drained sites
that were already forested before draining, or planted on a site specifically drained for forestry, are hereafter
referred to as 'forestry-drained', and 'afforested' is applied for sites that were previously used for other
purposes, in practice agricultural use or peat extraction.

## 2.2 Pre-processing

Data collection was done by assessing publications that provided either complete annual soil GHG balance
estimates or flux estimates with the potential for estimating annual soil GHG balance by using available
supplemental data (see Jauhiainen et al., 2019). Annual soil $CO_2$ balance estimates based on eddy covariance
flux monitoring and soil inventories (transformed from C values given in the original publications to $CO_2$)
were all added in the database without change.

The need for further processing to obtain an annual soil balance estimate was more common for $CO_2$ data than
for $CH_4$ and $N_2O$ data. The further processing of flux data was based on site-specific data or site-type-specific
data from the same climate zone, which were searched for in the literature, or obtained on our request from
either the authors or other specialists familiar with the conditions in the site or site type. Data were excluded
if specific enough supplemental data for estimating an annual soil balance was not found. The processing is
described in S1. The processing included, e.g., the incorporation of annual litter input and decomposition rates
in the estimates for studies where the ground surface in $CO_2$ flux monitoring points was kept free from litter
(9.2 % of the estimates in the boreal zone). If total soil $CO_2$ respiration was quantified, autotrophic root
respiration contribution in the total flux was proportioned by a coefficient to form an estimate of $CO_2$ emission
from decomposition (0.8 % of the estimates in the boreal zone and 78.8 % of the estimates in the temperate
zone). Some studies provided estimates of warm-season flux only (1.9 % of $CO_2$, 9.9 % of $CH_4$ and 0.6 % of
$N_2O$ estimates in the data), and we supplemented such data with cold-season flux by applying annualization
coefficients.

## 2.3 Structure of the database





While the IPCC (2014) EFs were based on 13 studies for $CO_2$, 23 for $CH_4$, and 20 for $N_2O$, we were able to increase the number of studies to 28 for $CO_2$, 33 for $CH_4$, and 32 for $N_2O$ (S1). Our search resulted in 595 annual soil GHG balance estimates: 210 about $CO_2$, 222 about $CH_4$ and 163 about $N_2O$ (Figure 1). Most of the $CO_2$ estimates were from boreal peatlands in Finland, and temperate zone data was mostly from Sweden. Of the $CO_2$ data, 49 annual soil balance estimates were based on soil C stock inventories and 161 on flux measurements (4 by eddy covariance and 157 by chamber measurements) (Table S1-1). In about 95 % of the studies the soil type was peat.

Each site in the database was identified using coordinates and site-type information from the original publications. Multiple-year GHG fluxes were available for most (55 % - 78 %) of the sites, and in these cases we recorded each annual GHG balance estimate as a single observation.

Site-specific data on climate, soil, and vegetation in the database were collected from the publications and used for defining the specific categories for which EFs were estimated. Methodology records include the monitoring method, number of spatial replicates, data collection year and period, and flux monitoring frequency. Management records include documented site management history (e.g., time since drainage, previous land use, fertilization applications, and information on planting and harvesting). Records on soil and hydrology include soil type, soil nutrient concentrations (C, N, P, C:N, P:N), pH, bulk density, peat depth, and WT characteristics. Records on site characteristics include general nutrient status description (ombrogenic vs. minerogenic, nutrient-poor vs. nutrient-rich), stand type, ground vegetation characteristics, and tree stand characteristics (stand type, stand-level basal area and stem volume, number of trees ha$^{-1}$). Temperature and precipitation conditions at the site locations for the measurement years ('weather') and over the previous 30 years ('climate') were also collected from the publications and, if needed, appended by data from the weather services closest to the site.

We categorized the monitoring sites in different ways, to evaluate how to use the data most efficiently for forming EFs. The most detailed categorization was done to boreal forestry-drained sites only, as they were represented by the clearly highest number of sites. The categories were based on climate, management history, nutrient status, ground vegetation, and forest productivity (Figure 2; Figure S2-1):

- Climate zone categories 'Boreal' and 'Temperate' based on FAO Climate/Vegetation Zones (Fig 4.1 in Volume 4, Chapter 4, IPCC, 2006).
- Soil type categories 'Peat' and 'Other organic soils'.
- Land management history categories 'Forestry-drained', Afforested from agricultural use, 'AF_AG', and Afforested from peat extraction, 'AF_PE'. Afforested sites were divided into these two categories since their soils may be considered quite different due to land-use legacies: in AF_AG generally nutrient rich and in AF_PE nutrient-poor, with likely differing soil GHG balances (e.g., Basiliko et al. 2007; IPCC 2014)
- Nutrient status categories (2 levels) 'Nutrient poor' ('NuP') and 'Nutrient rich' ('NuR') for forestry-drained sites, based on ombrogenic or minerogenic conditions prevailing before drainage, as reported in the publications or deducted from the information presented. In comparisons to IPCC (2014) EFs, Other organic soils and AF_AG sites were considered to be nutrient rich, and AF_PE sites nutrient poor.
- Nutrient status categories (4 levels) for forestry-drained boreal peatland sites, 'Extremely poor', 'Poor', 'Intermediate' and 'Rich'. This categorization was based on ground vegetation characteristics, for which nutrient status indicator information is available from classification studies and guidebooks (Päivänen and Hånell (2012)).
- Forest productivity categories 'Typical' and 'Low' for boreal forestry-drained sites, defined by combining information on site characteristics such as tree stand characteristics, soil nutrient status and





drainage conditions, and when floristically defined site type was available, also information on site type suitability for wood biomass production (Figure S2-1).

- o Sites were set into the 'Low' productivity category when they were poorly stocked (due to extremely low nutrient status, nutrient imbalance, or wetness despite ditches; e.g., Ojanen et al. 2019) but still fulfilled the minimum forest criteria as in FAO's FRA (2015). This was based on data and productivity studies from Finland, where most of the boreal sites were located (e.g., Laine et al., 2018).
- o These categories in nutrient-poor conditions were named 'Low_NuP' and 'Typical_NuP', and correspondingly in nutrient-rich conditions 'Low_NuR' and 'Typical_NuR'.
- Forestry-drained boreal peatland sites were further categorized based on their floristically defined site types that group together sites with similar ecology, soil and vegetation characteristics, following classifications presented by Päivänen and Hånell (2012). This information was commonly included in the publications in a manner that enabled consistent categorization (Figure 2; Figure S2-1). Sites
without such information in the publications were not forced into these categories, however.
- EFs were formed for all categories for which a minimum of 3 annual soil GHG balance estimates from different sites were available.

Further categories and continuous variables based on climate, site and vegetation characteristics were formed for evaluating their correlations with the EF estimates:
- Tree stand types formed the categories 'Conifer', 'Deciduous', and 'Mixed'.
- Based on ground vegetation characterized by shrubs or herbaceous plants, we formed categories describing ground vegetation 'shrubbyness': 'Yes' (shrubby) and 'No' (not shrubby). This was motivated by studies indicating that ericoid shrubs may suppress decomposition (Wang et al., 2015; Wiedermann et al., 2017).
- Soil C, N, P, C:N, P:N, bulk density and pH formed continuous variables.
- Tree stand variables basal area, stem volume, and number of trees formed continuous variables.
- Continuous weather and climate variables included annual air temperature, annual air temperature sum, mean air temperature in February, mean air temperature in July, annual precipitation, annual air temperature over 30 years, air temperature accumulated sum over 30 years, mean air temperature in
February over 30 years, mean air temperature in July over 30 years, annual precipitation over 30 years, site altitude from mean sea level, and site distance from the Arctic Circle.

For comparisons with the Tier 1 EFs in IPCC (2014), we divided our data into comparable categories. IPCC (2014) category for boreal zone 'Forest Land, drained, including shrubland and drained land that may not be
classified as forest and Nutrient-poor sites' (in Figure 3 referred to as FAO&NuP) equals our three categories pooled together ('Low_NuP', 'Typical_NuP', and 'AF_PE'), and 'Nutrient-poor sites' category in IPCC is comparable with our two categories pooled ('Typical_NuP', 'AF_PE'). Boreal zone 'Nutrient-rich sites' category in IPCC (2014) equals our three categories pooled ('Low_NuR', 'Typical_NuR', 'AF_AG'). The 'Temperate' category in IPCC (2014) includes all data found for the temperate zone, and is thus comparable
with pooled data from all our four temperate zone categories ('NuP', 'NuR', 'AF_AG', and 'Other organic soils').

### 2.4 Analyses

*EFs in differing category setups*



For forming the EFs, we calculated averages only for categories that included at least 3 soil GHG balance estimate representing different sites (i.e., at least 1 estimate from 3 sites in the category). To address differences in flux data composition based on chamber techniques, e.g. spatial coverage of data collection at the field sites and origin and quality of data types used for compiling the soil GHG balance estimate. we implemented 'relative data reliability weighting' by giving less reliable estimates a weight of 0.5 in selected analyses  denoted by 'weighted means' (see 'Relative data reliability weighting' in Supplement 1 for details). We inspected EFs in the following categories:

- $CO_2$, $CH_4$ and $N_2O$ EFs (weighted means) in categories that were the same as in IPCC 2014,
- $CO_2$, $CH_4$ and $N_2O$ EFs (weighted means) in different nutrient status and productivity categories in forestry drained and afforested sites, and in other organic soils,
- $CO_2$ EFs (weighted means) based on flux data and inventory data,
- $CO_2$, $CH_4$ and $N_2O$ EFs (arithmetic means) in forestry-drained peatlands in the boreal climate zone classified into specific site types and site nutrient status categories,

*Correlations with climate, soil, and vegetation variables*

The analysis of annual soil GHG balance correlations with environmetal characteristics was restricted to peat soils in forestry-drained sites in the boreal and temperate zones because it formed the largest dataset. This analysis included only GHG data collected by flux monitoring methods because many of the variables were year-specific (e.g. weather variables, tree stand basal area) and this selection made it possible to combine the annual soil GHG balances with the conditions of specific monitoring years .

In total 29 variables related to soil, vegetation, weather, or climate were tested for possible correlation with soil GHG balance estimates (Figure S2-2). Variables included in the analyses (reference category for categorical variables is underlined) were:

- Soil
    - Nutrient status (2 levels): 'Nutrient-poor' (NuP) vs. 'Nutrient-rich' (NuR)
    - Nutrient status (4 levels): 'Extremely poor', 'Poor', 'Intermediate', 'Rich'
    - Continuous variables: C, N, C:N, P, P:N, bulk density, pH
- Vegetation
    - Forest productivity: 'Typical' vs. 'Low'
    - Productivity and nutrient status: 'Low_NuP' vs. 'Typical NuP' vs. 'Low NuR' vs. 'Typical NuR'
    - Stand type: 'Conifer', 'Deciduous', 'Mixed'
    - Ground vegetation dominance by shrubs (shrubbyness): 'Yes' vs. 'No'
    - Continuous variables describing the tree stand: 'Basal area', 'Stand volume of trees', 'Number of trees'
- Weather, Climate
    - Climate zone: 'Boreal' vs. 'Temperate'
    - Continuous variables: 'Altitude', 'Distance from the Arctic Circle southwards', multiple annual and long-term average temperature and precipitation variables

The analysis was based on linear mixed models with *site* as a random effect. Data on $CH_4$ and $N_2O$ estimates were log-transformed (after adding a small constant to make all values positive) to make the model residuals more homoscedastic. Less reliable data (based on low number of spatial replicates, see 'Relative data reliability weighting' in S1 for details) points contributed to the model fits by half of the weight of others. Means of soil annual GHG balance by category were estimated using a simple linear mixed model with intercept only, site as random effect, and data restricted to the target category.





Univariate models were first fitted separately for each potential covariate. Multiple linear models were then
developed using stepwise regression with backward elimination starting from an initial model containing all
covariates that were significant in univariate models, except that:

- highly collinear covariates were avoided by choosing only one of "Nutrient status", either 'Nutrient
status' and 'Forest productivity' or their combination 'Productivity and nutrient status', and only one
of the continuous soil, temperature and precipitation variables,
- of vegetation variables defining tree stand, stand volume of trees was chosen, because the others were
provided for clearly fewer data points,
- climate zone was not used because a wider range of conditions became available for finding
correlations between the climate variables and soil GHG balance by using continuous climate
variables, and the dataset for the temperate zone was too small to allow within-zone analyses to be
done.

The choice between the alternative covariates was based on the Akaike information criterion (AIC) in
univariate models fitted to a subset of data for which all compared covariates were available. In each step of
backward elimination, a subset of data was used for which all current covariates were available, and the least
significant covariate was dropped, until all remaining covariates were significant ($p<0.05$). For variable
selection, continuous covariates were scaled to zero mean and unit variance, but the coefficients presented in
the result tables are associated with unscaled covariates.

## 3 Results

### 3.1 EF estimates for the IPCC (2014) Tier 1 EF categories

The EFs estimated in this study were generally in line with the EFs provided by IPCC (2014) (Figure 3).
However, $CO_2$ EFs in this study were consistently, but, based on the confidence intervals, not significantly,
lower than the EFs provided by IPCC. The $N_2O$ EFs were higher for boreal nutrient-poor forests (NuP) and
clearly smaller for the temperate zone than the EFs by IPCC, but also here the confidence intervals overlap.
The confidence intervals were mainly similar in this study and in IPCC (2014), except for a wider interval for
the temperate zone $CH_4$ EF, and smaller interval for the temperate zone $N_2O$ EF in this study.

### 3.2 EF estimates for more specific EF categories

#### 3.2.1 $CO_2$

The average $CO_2$ EFs for the boreal zone showed net emissions (positive flux numbers) in all categories
except an removal for the afforested peat extraction site category (AF_PE: -86 g m$^{-2}$ y$^{-1}$) (Figure 4; numeric
values are available in Table S2-3). The highest $CO_2$ EFs were in the low-productivity nutrient-poor (Low
NuP: 269 g m$^{-2}$ y$^{-1}$) and typical-productivity nutrient-rich (Typical NuR: 260 g m$^{-2}$ y$^{-1}$) categories using
inventory data. The individual soil $CO_2$ balance estimates behind the EFs included both negative and positive
values in all categories, and the lower 95% confidence limits were negative in all categories except the Typical
NuR category using flux or combined data.

For the temperate zone, the $CO_2$ EFs showed on average emissions in all categories, and also a great majority
of the soil $CO_2$ balance estimates were effluxes (Figure 4; Table S2-3). The EFs were highest and confidence
intervals widest in afforested agricultural lands (AF_AG: 932 g m$^{-2}$ y$^{-1}$) and 'other organic' soils (960 g m$^{-2}$
y$^{-1}$). Overall, the EFs were smaller in the boreal zone (maximum 269 g m$^{-2}$ y$^{-1}$) than in the temperate zone
(minimum 535 g m$^{-2}$ y$^{-1}$).



Boreal typical-productivity forestry-drained categories included enough estimates based on both flux and inventory data to allow comparison of EFs. Inventory data resulted in higher EF for nutrient-rich sites and flux data resulted in somewhat higher EF for nutrient-poor sites (Figure 4). The confidence intervals were overlapping, however. For the low-productivity categories there was not enough flux data for specific EFs, but the flux-data estimates were generally smaller than the inventory data estimates, as shown by EFs with pooled data (Figure 4).

The $CO_2$ EFs averaged for more detailed nutrient status categories were generally emissions to the atmosphere, except the EF for extremely nutrient-poor sites based on flux data (Figure 5; Table S2-2) whereas the median values were closer to zero and could indicate either emission or removal. Both mean and median $CO_2$ values were relatively similar between inventory and flux data in most categories, but a higher count of outlier values was more typical for the flux data. The most notable difference between measurement approaches was observed for low-productivity forests in the extremely nutrient-poor sites where inventory data resulted in high emission (mean 369 g m$^{-2}$ y$^{-1}$, median 495 g m$^{-2}$ y$^{-1}$), but the few available flux data estimates indicated removals or close to zero values (Figure 5).

### 3.2.2 $CH_4$

The highest $CH_4$ EF was observed for boreal low-productivity nutrient-poor forestry-drained peatlands (Low NuP; 2.48 g m$^{-2}$ y$^{-1}$), which was clearly higher than the EF for nutrient-poor sites in IPCC (2014) (Figure 6; Table S2-3). The EF for typical-productivity nutrient-poor category remained at the same level as the IPCC EF, while the EF for nutrient-rich forestry-drained peatlands was somewhat higher than the IPCC one. $CH_4$ EFs in the two boreal afforested site categories showed minor removals (from -0.36 to -0.63 g m$^{-2}$ y$^{-1}$) from the atmosphere, in contrast to IPCC EFs (Figure 6).

$CH_4$ EFs in the temperate zone showed emissions of ca. 1 g m$^{-2}$ y$^{-1}$ in both forestry-drained peatland categories (0.94, 1.03 g m$^{-2}$ y$^{-1}$), and close to zero (0.07 g m$^{-2}$ y$^{-1}$) in the other organic soils (Figure 6). The afforested agricultural lands in the temperate zone resulted on average $CH_4$ removals (-0.33 g m$^{-2}$ y$^{-1}$). In the temperate zone, EFs were higher than those by IPCC for forestry-drained peatlands and lower than IPCC default for afforested lands and for other organic soils.

When the effect of site nutrient status was examined with more detailed categories for the boreal zone, the extremely nutrient-poor sites showed the highest $CH_4$ EF, and the emissions decreased with increasing site nutrient status (Figure 7). All site categories had positive EFs (Figure 7; Table S2-3). Medians, on the other hand, indicated zero for intermediate sites, and removals for the most nutrient-rich sites (Figure 7). Outlier $CH_4$ data values on both sides of the median were found in all categories, but were more common in the intermediate and rich conditions (Figure 7).

### 3.2.3 $N_2O$

The highest $N_2O$ EF (1.38 g m$^{-2}$ y$^{-1}$) was found for afforested agricultural lands in the boreal zone (Figure 8; Table S2-3). $N_2O$ EF for the afforested peat extraction areas (0.35 g m$^{-2}$ y$^{-1}$) was similar to the typical-productivity nutrient-rich forestry-drained peatlands (0.34 g m$^{-2}$ y$^{-1}$), followed by low-productivity nutrient-rich (0.12 g m$^{-2}$ y$^{-1}$) and typical-productivity nutrient-poor (0.14 g m$^{-2}$ y$^{-1}$) forestry-drained peatlands. In the temperate zone, the highest $N_2O$ EFs were observed in the typical-productivity nutrient-rich forestry-drained peatlands (1.26 g m$^{-2}$ y$^{-1}$), followed by afforested agricultural lands (0.75 g m$^{-2}$ y$^{-1}$) (Figure 8). In the other organic soils category, the $N_2O$ EF was smaller (0.16 g m$^{-2}$ y$^{-1}$). Confidence intervals were wide in the temperate zone nutrient-rich sites, and in the afforested agricultural sites in both climate zones.





N$_2$O EFs in forestry-drained boreal peatlands showed mostly emissions from the soil but the boxplots show that most of the soil N$_2$O balance estimates are actually close to zero (Figure 9). The outlier values, mostly in intermediate and rich sites, result in wide confidence intervals and have a major effect on the average as they were typically sites with very high emissions. Overall, the data distribution in the intermediate category, especially, was highly skewed towards high positive values.

**3.3 Correlations with weather and climate variables and site type characteristics**

Several variables related to soil, vegetation, weather of the monitoring year and climate were significantly correlated (p-value ≤ 0.05) with soil GHG balance estimates (Table S2-4). Based on the univariate models, the annual soil CO$_2$ balance was negatively correlated with soil C concentration (i.e., emission decreased with increasing soil C concentration; note that these were all high-C peat soils, where lower C generally indicates more nutrient-rich conditions) and C:N ratio, and positively correlated with soil bulk density and several

variables related to long warm-season conditions and elevated temperatures (Table 2). Of vegetation-related variables, only the stand type correlated with the annual soil CO$_2$ balance. In comparison to conifer stands, deciduous stands showed higher and mixed tree stands smaller soil CO$_2$ emissions (Table 2). The best multiple model explained 41% of the variation with soil C:N, stand type, and mean temperature over 30 years as explaining covariates (Table 3). In the univariate model, deciduous stands affected the soil CO$_2$ balance more

than mixed stands, whereas in the multiple model it was vice versa. This indicates some interaction between stand type, soil characteristics and climatic conditions.

The annual soil CH$_4$ emissions were higher for nutrient-poor sites than for nutrient-rich sites (Table 2, Table 3). The higher potential for CH$_4$ emissions on nutrient-poor sites was further supported by vegetation-related

predictors such as low productivity, low stand volume of trees and the low number of trees. Site nutrient status and productivity are partly correlated, which explains the differing patterns found for these predictors in multiple versus univariate models. Further, CH$_4$ emissions were negatively related to temperature, i.e. colder conditions resulted lower efflux. The best multiple model explained only 28% of the variation and was a combination of the variables site nutrient status, site productivity class, and February mean temperature (Table

3).

The annual emission of N$_2$O correlated positively with soil bulk density, N concentration, and pH (Table 2). Further, a positive correlation was found with vegetation predictors describing the size and density of the tree stand (basal area, stem number and stand volume), and emission was higher in mixed stands in comparison to conifer stands (Table 2, Table 3). Of weather and climate variables, the annual soil N$_2$O balance correlated

positively with indicators of warmer conditions (annual and/or long-term warm season temperatures, annual mean temperature, and southern location), and also with annual precipitation. Soil N concentration, stand type, and July mean temperature over 30 years were combined in the best multiple model which was able to explain 51% of the variation (Table 3). It should be noted that the multiple models presented here were built with variables that were most widely available in the dataset. With smaller datasets for which a wider set of

variables were available, higher explanatory power could be obtained. For example, with two vegetation-related covariates (stand type and stand volume of trees) and soil pH as much as 83% of the N$_2$O emissions could be explained, but the number of observations was only 21 (Table S2-5).

**4 Discussion**

**4.1 Success in using the data for modeling and creating categories for specific conditions**





Since the release of the IPCC report (2014), there have been many new publications reporting annual soil GHG balances. These data are enough for constructing more site specific EFs as our results show. Yet, as we pointed out in an earlier study (Jauhiainen et al., 2019) there has been little consistency in including environmental data in the publications that would facilitate efficient re-use and pooling of data. Consequently, the effort to identify the major drivers of the variation in soil GHG balance (and EFs) from the potential

predictors describing soil, vegetation and climate characteristics may not have led to optimal choices, as we were limited to variables that were at least somewhat consistently reported. In this study, the predictive power of the "best" multivariate models was reasonable, ranging from 28% for $CH_4$ to 51% for $N_2O$. As expected, variables describing soil nutrient status, both continuous and categorical, were significantly related to the soil GHG balances, and an increase in soil nutrient status led to higher emissions of $CO_2$ and $N_2O$ but lower $CH_4$

emissions or even switch into a small sink. Overall, the responses of $CO_2$ and $N_2O$ often go hand in hand in the opposite direction compared to $CH_4$. This may be explained by the WT that, unfortunately, was not so generally available in the publications that it could have been used in the models (Jauhiainen et al., 2019). A deeper WT will increase the extent of the oxic surface soil layer where $CH_4$ oxidation takes place and thus reduces the emission (Minkkinen and Laine 2006; Ojanen et al., 2010; Jauhiainen et al., 2019; Rey-Sanchez

et al., 2019), while it also allows for more efficient aerobic decomposition that leads to a higher $CO_2$ emission (e.g., Jaatinen et al., 2008; Ojanen and Minkkinen, 2019) and also to some extent, although not linearly, favours processes that increase $N_2O$ emission (Pärn et al., 2018). The relatively low power of the $CH_4$ model may be due to the sources and sinks all being so small in drained forests that their overall variation was clearly smaller than that in the explanatory variables. On the other hand, strong dependence of the annual soil $CH_4$

balance on both WT and tree stand volume has been observed in earlier studies (Minkkinen et al., 2007; ). Tree stand volume and the depth of the WT are generally positively correlated, which is most probably the causal reason for the relationship. The relationship was not well present in this pooled data, possibly due to the skewed distribution of the data that the log-transformation may not have fully remedied.

The WT is generally deeper in nutrient-rich sites that are more densely stocked (Ojanen et al., 2013), in warmer

climatic conditions where evapotranspiration is higher, and under deciduous trees that show higher evapotranspiration than conifers (Leppä et al., 2020). This was well reflected in the models. Tree stand characteristics, including the stand type (conifer, deciduous, mixed), are thus potentially very useful for predicting emissions as in peatlands they reflect both the site nutrient status and WT regime (e.g. Laine et al., 2004). Litter quality further differs between the stand types (Preston et al., 2000; Cornwell et al., 2008), which

may explain the higher $CO_2$ emissions from deciduous stands that generally produce more readily decomposable litter than coniferous stands. However, why $N_2O$ emissions were highest in mixed stands remains unexplained. It must be noted that the literature data were solely from stands that had not been recently managed, and short-term management impacts need to be studied separately.

In addition to the modelling efforts, the data allowed defining new site categories that distinguish more specific

environmental conditions than the IPCC (2014) Tier 1 categories. These included other organic soils than peat, afforested sites with different land-use histories (agriculture versus peat extraction), and more detailed categories for forestry-drained sites based on site nutrient status and forest production potential. This resulted in narrower confidence intervals in most categories than in the IPCC (2014) Tier 1 categories. The number of estimates included in each category was typically smaller when applying more categories; however, this did

not necessarily lead to higher uncertainty as shown by our results.

The overall soil nutrient status and potential forest productivity can be largely interpreted from the vegetation community characteristics (Päivänen and Hånell, 2012), based on which more specific categories than the very broad "poor" and "rich" can be distinguished. The two broad categories as used in IPCC (2014) describe in a broad sense ombrogenic versus minerogenic groups. Within minerogenic sites, especially, there is,




however, high variation from rather nutrient-poor (oligotrophic) to mesotrophic (intermediate) and eutrophic (nutrient-rich) conditions. Within ombrogenic sites, there is also variation from extremely poor sites not at all suitable for forest production because of very low productivity to pine-dominated sites where forest growth may be satisfactory from the forestry point of view (Keltikangas et al., 1986). Within both poor and rich sites, there are also low-productivity variants, i.e., sites where tree growth remains poor due to overall scarcity of

nutrients, especially N, or due to strong imbalance of scarce other nutrients versus high N that is typical for drained sites that were originally very wet treeless mires (Moilanen et al., 2010; Ojanen et al., 2019). Such conditions can be recognized and classified based on expert information found in site-type classification and forest productivity research reports (e.g., Keltikangas et al., 1986) and described in forestry-oriented site-type guidebooks (e.g., Laine et al., 2018; Bušs, 1981). These categories can naturally only be applied if such

classification information is available. A challenge is that such national systems of site classification are difficult to combine into consistent international categories.

### 4.2 EFs in differing category setups

By using the categories applied in the IPCC (2014), the EFs estimated in this assessment were relatively similar to those provided by the IPCC. The values for the temperate zone changed the most due to the relatively

largest addition of new data. In the more site-specific analysis, the annual soil $CO_2$ balances in afforested sites were, firstly, expected to differ from those in forestry-drained sites due to the legacy effects of the preceding land management. In the temperate zone, the two afforested site categories, and also the 'other organic soils' category represented the high end of annual soil $CO_2$ balances. Data from afforested and 'other organic soils' sites formed likely the high-end estimates in the only EF category available for the temperate zone in IPCC

(2014). In the boreal zone the afforested site soil $CO_2$ balances are closer to zero than the averages in forestry drained sites. The modest amount of WT data available for this assessment suggests that afforested and other organic soils category sites have in general deeper WT than forestry-drained sites (Table S2-6). This would logically contribute to both higher $CO_2$ emissions and lower $CH_4$ emissions than in forestry-drained sites. However, in boreal data the afforested sites had lower CO2 EFs than forestry-drained sites, though with very

wide confidence intervals. For many of these sites, we had to introduce litter inputs and their decomposition, which adds uncertainty in the estimates. Dividing the data into separate afforested site categories resulted in their annual $CH_4$ balances being small sinks while forestry-drained peatland sites remained on average as emission sources. The other organic soils in the temperate zone showed a consistently close-to-zero annual soil $CH_4$ balance.

Concerning $N_2O$, the EF of afforested agricultural land in the boreal zone is quite striking, showing higher emissions than in any other category, being also higher than the EFs in the IPCC (2014). We have no immediate explanation for this. In the temperate zone, $N_2O$ EFs and confidence intervals in the three categories were narrower in this study compared to that in the single category available in the IPCC (2014). The other organic soils in the temperate zone showed relatively low $N_2O$ emission, which together with the EFs of the

other two gases suggests that their GHG fluxes differ from peat soils.

The forestry-drained sites in the boreal zone had the most extensive and consistent data, and allowed further separation of nutrient status categories and typical-productivity versus low-productivity sites. In IPCC (2014) lower-than-typical productivity was distinguished for nutrient-poor boreal sites so that EFs were estimated separately for typical-productivity sites and combined typical and low-productivity sites (Table 1). In this

study, average soil $CO_2$ balances in low-productivity conditions resulted in emissions in all site nutrient status groups from extremely poor to rich conditions (Figures 4, 5). In intermediate and rich site conditions the differences between low-productivity and typical-productivity $CO_2$ emissions were less notable in comparison to the large differences found between poor and extremely poor site conditions. This was unexpected, as earlier



research has indicated low decomposition rates for litters and peat formed in nutrient-poor conditions (e.g.,
Hogg et al., 1992; Straková et al., 2012; Harris et al., 2020), where decomposers are further limited by low
nutrient availability (Bragazza et al., 2007). It may be noted that the Low NuP category was almost entirely
represented by inventory data, but how that could produce an artifact especially in extremely nutrient-poor
sites is not explained by the data.

The Low NuP category further showed the highest average $CH_4$ EF among the boreal zone categories. This
can be explained by the higher WT in forest stands of low density (Table S2-6; Minkkinen et al., 2007). The
$CH_4$ EF in the Low NuP category is considerably higher than the EFs in the IPCC (2014) assessment where
the NuP category includes data both from typical- and low-productivity sites. Among the three GHGs studied,
$CH_4$ EFs varied most consistently along the nutrient status gradient with the highest emissions in the poorest
nutrient status environments characterized by ground lichens, and smaller emissions towards increasingly
nutrient-rich – and drier due to higher tree stand evapotranspiration – environments. This decreasing trend in
emissions was clear in both in average and median values, and visible also in annual averages by site types
(available in Figure S2-3).

Most of the $N_2O$ data was available from N-rich conditions (see Table S2-6) likely because of higher interest
to study these environments for their large emission potential. The $N_2O$ EFs increased from extremely poor to
intermediate environments in forestry-drained peatlands. Soil $N_2O$ balance means, medians, and data
distribution in the boxplots show that extremely poor and poor sites have smaller emissions than intermediate
and rich conditions where a higher number of extreme values were also observed. In this study, especially
sites classified as intermediate showed high emissions. We have no explanation for this that could be based,
e.g., on site type classification as presented in the original publications and what is known about peat N
concentrations in different site type classes (e.g., Westman and Laiho, 2003). Such sites with extreme values
should be considered as locations for further research potentially increasing the understanding of GHG flux
formation in different environments. If consistent patterns explaining the high emissions were found, specific
categories could then be formed for such sites.

### 4.3 $CO_2$ balance estimates based on inventory versus flux data and means versus medians

$CO_2$ emissions proved to be the most complex fluxes because the flux monitoring and inventory methods use
profoundly different approaches (Jauhiainen et al., 2019). The available data structure was not optimal for
comparisons between the two methods because only two of the formed categories included enough data
produced by both methods. In those categories, the difference in the resulting EF was small considering the
differences in the methods and that the data were from different sites.

It remains to be further clarified why inventory data resulted in high C losses from the extremely poor sites,
whereas the flux data indicated close to zero emission or a C sink for these sites. If methods were distributed
to distinctly different site types this could introduce a bias; the pre-drainage peat in dryish extremely poor sites
largely consists of such hummock species, e.g., *Sphagnum fuscum*, that are generally considered decay
resistant, while the remains of species typical of originally wet extremely poor sites, e.g., *S. cuspidatum*,
decompose more readily (e.g., Bengtsson et al., 2018). Consequently, if sampling focused on different site
groups, different results could be expected; however, such a difference between methods was not evident from
the site descriptions. Further, in extremely poor sites that are generally at late stages of autogenic peatland
succession before drainage, deep horizons usually represent different conditions and plant communities that
are typically less decay resistant and could contribute to C loss over longer periods captured by the inventory
methods, but the drainage effect is unlikely to extend very deep in these sites. Thus, we can not explain the
difference.



Although the average soil $CO_2$ balance values in poor, intermediate and rich site groups resulted in emissions for both flux and inventory data, median values indicated variably emissions or a sink closer to zero. Several extreme soil $CO_2$ balance values in flux data widen the deviation around the means, which may be an outcome either from differing environmental conditions resulting in widely different values in the groups, temporal scale differences of the methods (the temporal scale of flux measurements is annual whereas for stock inventory it is decadal – given the importance of, for example, climate, more variability in estimates with flux measurements is expected), or unidentified quality issues in some of the data values.

Both mean and median values may be used for describing GHG flux data, the mean being the more common descriptor, e.g., in IPCC (2014), and in this assessment only 2 publications out of 54 reported median values. Normal distribution of the data is expected when using means, while medians would be best to use for skewed data. The distributions of monitored data were typically not provided, and thus we were not able to assess the normality criterium. In the pooled data, mean EFs were somewhat higher than the corresponding median values in different nutrient status categories (Figures 5, 7, 9), which shows the impact of positive extreme values, shown by the boxplots, on means. The medians likely better describe typical conditions. Consequently, we suggest that medians could in fact be a useful alternative for forming the EFs.

### 4.4 Data issues

A notable number of new studies have become available since the IPCC (2014) assessment, which allowed including more specific conditions than the current IPCC Tier 1 categories, and to inspect the data structure. The number of estimates in several categories differed in the two assessments, which is likely a combined outcome of a larger database available in this study and also differences in accounting for individual sites. Direct database structure and numeric estimate comparisons between these two assessments cannot be evaluated in detail due to limited access to the IPCC (2014) data. Differences in data processing in the two assessments do exist. As an example, differences in the number of sites in the EF table (see Table 1 and Figure 2) in the IPCC 2014 assessment and this study cannot be readily explained by new studies becoming available as IPCC shows higher number of measurements. In this study, each site was identified based on coordinates and reported site characteristics irrespective of the publication where the data was presented, which may differ from the method applied in IPCC (2014). This would mean that data from a site reported in different papers may have been used as different independent observations in the IPCC assessment.

Obvious limitations in building a database from published data are the certain randomness of the site types and conditions included in the publications, and the inconsistent reporting of field conditions (e.g. site types, tree stand and soil characteristics, collected data types; Jauhiainen et al., 2019). In this study, the generally available information included land use history, soil type (peat, other organic soils), and site nutrient status information. The number of estimates falling into specific data categories could not be controlled, but a lower limit of estimates qualifying inclusion to analysis (in this study n≥3 is used) could be set. Differences in data collection procedures in the original studies had to be accepted, likely resulting in inconsistencies in data quality.

Major sources of inconsistencies in the data, reviewed by Jauhiainen et al., (2019), included (a) a large proportion of $CO_2$ flux estimates based on day-time flux data, not assessing the impact of generally cooler night-time periods, (b) potential biases in $CO_2$ flux estimates in studies not specifying whether ground vegetation efflux were included in the flux monitoring, and (c) some studies using a generic literature-based proportion (50 %) for autotrophic root respiration to modify the total $CO_2$ flux monitored that is not necessarily representative for conditions on the site (e.g. tree stock, climate). The existence of the above-listed inconsistencies was recognized also in this analysis. Consequently, modifications aiming at improvements in





the data structure and the use of data reliability weighting were applied to increase data consistency (see S1). However, analyses of strengths and weaknesses in individual data sources were surely out of the scope of this study.

While we were synthesising the existing data, still more data has become available, e.g., Sosulski et al. (2019), Ernfors et al. (2020), Bjarnadottir et al. (2021), Butlers et al. (2021), Jovani-Sancho et al. (2021), and Hermans et al. (2022). All new data may be added to our original database that is openly available at (… data repository to be added).

**5 Conclusions**

This study added considerably to the number of publications applicable for forming emission factors for forests on drained organic soils in the boreal and temperate zones. When using the broad EF categories of the IPCC (2014) assessment, the added data caused only modest changes in the estimated EFs and their confidence intervals, indicating overall good compatibility of our work with the IPCC assessment. More specific site-type categories in our assessment generally resulted in narrower confidence interval around the category average compared to the present Tier 1 categories (IPCC 2014), which supports, e.g., the use of different categories for afforested sites and forestry-drained sites, and more specific site nutrient-status and productivity categories based on timber production potential. We found a strong negative relationship between annual soil $CH_4$ balance and site nutrient-status gradient, a relatively clear trend of increase in $N_2O$ emissions over increasing site nutrient-status gradient, while the patterns in $CO_2$ balance were more variable. Occasional wide confidence interval around the mean EF resulted more typically from single or few highly deviating estimates, rather than because of a wide range in conditions within the categories applied. Despite variably available supporting data in the publications, statistical analyses supported these findings by connecting the annual soil GHG balances to site-specific soil nutrient status indicators, tree stand characteristics and temperature-associated weather and climate variables. Although the flux and inventory methods are profoundly different in soil C balance monitoring, the EFs and their uncertainties did not differ much when estimated for similar environments with comparable data. The most obvious data need is in the temperate zone with regard to all site categories. Another data need relates generally to reporting of background data for the monitoring sites, both on environmental conditions (e.g., WT characteristics over years) and tree stand descriptors, as both were found important for explaining the variation in the annual soil GHG balances. Finally, as soil characteristics especially in peatlands evolve when more time passes since drainage, time series of explanatory and categorical variables, as well as litter input and decomposition dynamics, are needed.

*Data availability*. The research data used in this study are peer reviewed publications listed in Supplement S1. The main data content is made available in data repository at: link here.

*Supplements*. The supplements related to this article are available online at: doi-link here.

*Author contributions*. All authors planned the research jointly and contributed to data collection. JJ retrieved and reviewed the data, compiled the database, wrote the first draft of the manuscript, compiled supplementary information, coordinated the commenting and revisions that were provided by all authors. RL, JJ and JH planned content for statistical analyses correlating GHGs with climate variables and environmental variables, and JH conducted the analyses. JJ compiled the following versions together with RL and JH.

*Competing interests*. The authors declare that they have no conflict of interest.



*Acknowledgements.* This study was part of the SNS-120 project 'Anthropogenic greenhouse gas emissions from organic forest soils: improved inventories and implications for sustainable management' funded by Nordic Forest Research (SNS) with in-kind funding from all project partners (The Norwegian Environmental Agency for Norway, Nibio). For refining into publication this study was supported by the
685 'Demonstration of climate change mitigation measures in nutrients rich drained organic soils in Baltic States and Finland' (LIFE OrgBalt, LIFE18 CCM/LV/001158). This study was further supported by the Academy of Finland (grant 289116), Ministry of Education and Research of Estonia (grant PRG-352), Danish Innovation Fund for the Eranet Facce-Eragas project INVENT (grant 7108-00003b; FACCE ERA-GAS has received funding from the European Union's Horizon 2020 research and innovation programme under grant
agreement No 696356), University of Helsinki grant to 'Peatland Ecology Group', and the European Union through the Centre of Excellence EcolChange in Estonia.

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





**Table 1. IPCC (2014) Tier 1 CO₂, CH₄ and N₂O emission factors (EFs) for boreal and temperate drained organic forest soils, as average (Ave), uncertainty (95% confidence limits, CI), and number of observations (i.e. number of sites) in the category (N).**

| Forest site type and climate zone | EF $CO_2$ (g m$^{-2}$ y$^{-1}$) [2] | | | EF $CH_4$ (g m$^{-2}$ y$^{-1}$) [2] | | | EF $N_2O$ (g m$^{-2}$ y$^{-1}$) [2] | | |
|---|---|---|---|---|---|---|---|---|---|
| | Ave | 95% CI | N | Ave | 95% CI | N | Ave | 95% CI | N |
| Forest land, drained, including shrubland and drained land that may not be classified as forest[1] and nutrient-poor sites in boreal zone | 135.8 | −40.4 – 308.3 | 63 | − | | | − | | |
| Nutrient-rich sites in boreal zone | 341.3 | 198.2 – 477.1 | 62 | 0.20 | −0.16 – 0.55 | 83 | 0.503 | 0.299 – 0.707 | 75 |
| Nutrient-poor sites in boreal zone | 91.8 | −84.4 – 267.9 | 59 | 0.70 | 0.29 – 1.10 | 47 | 0.035 | 0.024 – 0.044 | 43 |
| All sites in temperate zone | 954.2 | 734.0 – 1211.1 | 8 | 0.25 | −0.06 – 0.57 | 13 | 0.440 | −0.090 – 0.959 | 13 |

[1] Sites with poor tree growth (due to extremely low nutrient availability, nutrient imbalance or wetness, but still fulfilling the minimum criteria as in FAO's Forest Resources Assessment (FRA, 2015).
[2] Values are converted from Tables 2.1, 2.3, and 2.5 in IPCC (2014), where the unit for $CO_2$ is 'tons $CO_2$-C ha$^{-1}$ y$^{-1}$' and for $CH_4$ and $N_2O$ 'kg ha$^{-1}$ y$^{-1}$'.



**Table 2. Parameter estimates with standard errors (SE) for significant covariates in univariate models of annual soil GHG balances. For categorical covariates, the number of parameters is one less than the number of categories and they can be interpreted as differences from the reference category for which no estimate is provided. The p-values for the significance of the difference of the parameter estimate from 0 are based on Wald tests. The numbers of data points (n) and sites (n_sites) used for each model are given.**

| Covariate | Unit / category | Estimate | SE | p | n | n_sites |
|---|---|---|---|---|---|---|
| **CO_2** | | | | | | |
| Soil C | % | -36.66 | 10.25 | 0.001 | 30 | 28 |
| Soil C:N | % | -15.96 | 4.10 | 0.000 | 100 | 93 |
| Soil bulk density | g cm$^{-3}$ | 521.9 | 257.3 | 0.045 | 112 | 98 |
| Stand type | Conifer[a] | 0 | | | 134 | 118 |
| | Deciduous | 621.9 | 117.1 | 0.000 | | |
| | Mixed | -111.5 | 95.8 | 0.249 | | |
| Climate zone | Boreal[a] | 0 | | | 134 | 118 |
| | Temperate | 423.0 | 127.5 | 0.001 | | |
| Southward distance from the Arctic Circle | km | 0.494 | 0.161 | 0.003 | | |
| Mean temperature of the measurement year | ºC | 51.73 | 24.83 | 0.039 | 127 | 113 |
| Temperature sum | degree days | 0.540 | 0.188 | 0.006 | 111 | 98 |
| July mean temperature | ºC | 45.42 | 18.91 | 0.020 | 130 | 115 |
| Mean temperature over 30 years | ºC | 79.79 | 27.97 | 0.005 | 134 | 118 |
| February mean temperature over 30 years | ºC | 45.73 | 17.22 | 0.009 | 134 | 118 |
| Mean precipitation over 30 years | mm y$^{-1}$ | 1.142 | 0.498 | 0.024 | 134 | 118 |

| Covariate | Unit / category | Estimate | SE | p | n | n_sites |
|---|---|---|---|---|---|---|
| **log(CH_4+ε)** | | | | | | |
| Nutrient status (2 levels) | NuP[a] | 0 | | | 188 | 124 |
| | NuR | -0.442 | 0.150 | 0.004 | | |
| Nutrient status (4 levels) | Intermediate[a] | 0 | | | 154 | 102 |
| | Extremely poor | 1.057 | 0.412 | 0.013 | | |
| | Poor | 0.335 | 0.201 | 0.099 | | |
| | Rich | -0.159 | 0.168 | 0.347 | | |
| Soil N | % | -0.460 | 0.104 | 0.000 | 68 | 46 |
| Soil C:N | | 0.015 | 0.007 | 0.040 | 131 | 95 |
| Soil P | mg kg$^{-1}$ | -0.001 | 0.000 | 0.025 | 39 | 26 |
| Site productivity class | Typical[a] | 0 | | | 188 | 124 |
| | Low | 1.030 | 0.320 | 0.002 | | |
| Productivity and nutrient status | Typical NuP[a] | 0 | | | 154 | 102 |
| | Low NuP | 0.782 | 0.413 | 0.062 | | |
| | Typical NuR | -0.392 | 0.168 | 0.022 | | |
| | Low NuR | 0.788 | 0.535 | 0.144 | | |
| Basal area of trees | m$^2$ ha$^{-1}$ | -0.020 | 0.006 | 0.011 | 15 | 7 |
| Stand volume of trees | m$^3$ ha$^{-1}$ | -0.004 | 0.001 | 0.000 | 140 | 101 |
| Stem number of trees | stems ha$^{-1}$ | -0.001 | 0.000 | 0.002 | 17 | 10 |
| Altitude | m | 0.004 | 0.001 | 0.002 | 188 | 124 |
| Mean temperature of the measurement year | ºC | -0.101 | 0.039 | 0.010 | 179 | 118 |
| Temperature sum | degree days | -0.002 | 0.001 | 0.006 | 163 | 108 |
| February mean temperature | ºC | -0.034 | 0.013 | 0.008 | 184 | 121 |
| July mean temperature | ºC | -0.167 | 0.067 | 0.014 | 188 | 124 |

| Covariate | Unit / category | Estimate | SE | p | n | n_sites |
|---|---|---|---|---|---|---|
| **log(N_2O+ε)** | | | | | | |
| Soil N | % | 0.231 | 0.087 | 0.011 | 58 | 40 |





| | | | | | | |
|---|---|---|---|---|---|---|
| Soil bulk density | g cm$^{-3}$ | | 0.886 | 0.288 | 0.005 | 67 | 50 |
| Soil pH | | | 0.382 | 0.177 | 0.037 | 44 | 27 |
| Stand type | Conifer[a] | 0 | | | | 90 | 67 |
| | Deciduous | | 0.143 | 0.115 | 0.219 | | |
| | Mixed | | 0.289 | 0.082 | 0.001 | | |
| Basal area of trees | m$^2$ ha$^{-1}$ | | 0.0080 | 0.0002 | 0.000 | 14 | 7 |
| Stand volume of trees | m$^3$ ha$^{-1}$ | | 0.0010 | 0.0004 | 0.029 | 67 | 52 |
| Stem number of trees | stems ha$^{-1}$ | | 0.0003 | 0.0000 | 0.000 | 13 | 6 |
| Southward distance from the Arctic Circle | km | | 0.0004 | 0.0001 | 0.000 | 99 | 73 |
| July mean temperature during the measurement year | ºC | | 0.048 | 0.024 | 0.049 | 95 | 70 |
| Mean temperature over 30 years | ºC | | 0.067 | 0.023 | 0.004 | 99 | 73 |
| February mean temperature over 30 years | ºC | | 0.031 | 0.013 | 0.024 | 99 | 73 |
| July mean temperature over 30 years | ºC | | 0.238 | 0.046 | 0.000 | 99 | 73 |
| Precipitation during the measurement year | mm y$^{-1}$ | | 0.0009 | 0.0002 | 0.000 | 95 | 70 |
| Mean precipitation over 30 years | mm y$^{-1}$ | | 0.0011 | 0.0003 | 0.000 | 99 | 73 |

[a] The reference category



**Table 3. Parameter estimates with standard errors (SE) and coefficient of determination (pseudo-$R^2$) in multiple linear models obtained by stepwise regression (see caption of Table 2 for details).**

| Predictor | Unit / category | Estimate | SE | p | n | $n_{sites}$ | $R^2$ |
|---|---|---|---|---|---|---|---|
| **$CO_2$** | | | | | | | 0.41 |
| Soil C:N | | -17.75 | 3.52 | 0.000 | 100 | 93 | |
| Stand type | Conifer[a] | 0 | | | | | |
| | Deciduous | 14.00 | 102.0 | 0.891 | | | |
| | Mixed | -182.2 | 53.6 | 0.004 | | | |
| Mean temperature over 30 years | °C | 108.2 | 19.6 | 0.000 | | | |
| | | | | | | | |
| **$\log(CH_4+\varepsilon)$** | | | | | | | 0.28 |
| Nutrient status | Intermediate [a] | 0 | | | 150 | 99 | |
| | Extremely poor | -0.085 | 0.588 | 0.886 | | | |
| | Poor | 0.435 | 0.180 | 0.017 | | | |
| | Rich | -0.115 | 0.153 | 0.454 | | | |
| Site productivity class | Typical[a] | 0 | | | | | |
| | Low | 1.065 | 0.473 | 0.027 | | | |
| February mean temperature | °C | -0.046 | 0.014 | 0.001 | | | |
| | | | | | | | |
| **$\log(N_2O+\varepsilon)$** | | | | | | | 0.51 |
| Soil N | % | 0.184 | 0.074 | 0.017 | 52 | 35 | |
| Stand type | Conifer[a] | 0 | | | | | |
| | Deciduous | -0.010 | 0.171 | 0.956 | | | |
| | Mixed | 0.396 | 0.114 | 0.002 | | | |
| July mean temperature over 30 years | °C | 0.297 | 0.062 | 0.000 | | | |

[a] The reference category





**Figure captions**

**Figure 1:** Monitoring sites providing annual soil GHG balance estimates for drained organic forest soils (red=peat, white=other organic soils) in the boreal and temperate zones. 'Estimates' show how many different annual estimates were available from the sites.

**Figure 2:** The categories for which EFs were estimated.

**Figure 3:** Emission factors (EFs) and their 95% confidence intervals for drained organic forest soils in categories used in IPCC (2014) for boreal and temperate zones: comparison of weighted means EFs obtained in this study and EFs in IPCC (2014). Number of sites providing soil annual GHG balance estimates (i.e. number of observations) from which the EFs were estimated are below the bars. Categories are explained in Figure 2 and Table S2-1, and numeric EF values from IPCC (2014) are in Table 1 and 975 values from this study are in Table S2-2.

**Figure 4:** $CO_2$ EFs (weighted mean ± 95% confidence intervals, n) in an expanded set of categories for the boreal and temperate zones. Averages based on only flux data, only inventory data and combined data are shown when applicable (n≥3). The dotted lines represent IPCC (2014) Tier 1 EF levels for categories including comparable data. Numbers below the bars give the numbers of observations from which the EFs 980 were estimated. Categories are explained in Figure 2 and Table S2-1, and values are in Table S2-3.

**Figure 5:** $CO_2$ EFs for four nutrient status categories further divided into low and typical productivity of boreal zone forestry-drained peatlands. The estimates are shown as the arithmetic mean ± 95% confidence intervals (left) and as boxplots (right), and separately for flux and inventory data when applicable (n≥3). In the boxplots, the central line inside the bar shows the median, the bottom and top box lines show the first 985 and third quartiles, respectively, the whiskers show the maximum and minimum values, and the circles represent outliers. Numbers below the bars (left) give the numbers of observations from which the EFs were estimated (the same for both figures), and numbers next to outliers in the boxplots (right) refer to publications listed in Table S1-1. The categories are explained in Figure 2, in Figure S2-1, and Table S2-1.

**Figure 6:** $CH_4$, weighted mean ± 95% confidence intervals) for drained organic forest soils in the boreal and 990 temperate zones. Category Low NuR is not included as it had <3 observations. The dotted lines represent IPCC (2014) Tier 1 EF levels for categories including comparable data. Categories are explained in Figure 2 and Table S2-1, and values are in Table S2-3.

**Figure 7:** $CH_4$ Efs for four nutrient status categories further divided into low and typical productivity of boreal zone forestry-drained peatlands, as the arithmetic mean ± 95% confidence intervals (left) and as the 995 median in boxplots (right). Boxplot characteristics are explained in Figure 5. Numbers below the bars (left) give the number of observations in each category, and numbers next to outliers in the boxplots (right) refer to publications listed in Table S1-1. The categories are explained in Figure 2, in Figure S2-1, and Table S2-1.

**Figure 8:** $N_2O$ EFs (weighted mean ± 95% confidence intervals) for drained organic forest soils in the 1000 boreal and temperate zones. Only categories with n≥ 3 included. The dotted lines represent IPCC (2014) Tier 1 EF levels for categories including comparable data. The categories are explained in Figure 2, in Figure S2-1, , and values are in Table S2-3.

**Figure 9:** $N_2O$ EFs for four nutrient status categories further divided into low and typical productivity of boreal forestry-drained peatlands, as the arithmetic mean ± 95% confidence intervals (left) and as the 1005 median in boxplots (right). Boxplot characteristics are explained in Figure 5. Numbers below the bars (left) give the number of observations in each category, and numbers next to outliers in the boxplots (right) refer to publications listed in Table S1-1. The categories are explained in Figure 2, in Figure S2-1, and Table S2-1.




**fig01**

| Country code | CO$_2$ | | CH$_4$ | | N$_2$O | |
|---|---|---|---|---|---|---|
| | Esti-mates | Sites | Esti-mates | Sites | Esti-mates | Sites |
| CAN | | | 3 | 1 | | |
| DE | 1 | 1 | 1 | 1 | 4 | 4 |
| DK | | | 1 | 1 | 1 | 1 |
| EST | 9 | 4 | 3 | 2 | 3 | 2 |
| FI | 175 | 127 | 176 | 94 | 113 | 57 |
| LV | 1 | 1 | | | | |
| SE | 18 | 4 | 30 | 4 | 36 | 3 |
| SL | | | 1 | 1 | 1 | 1 |
| UK | 6 | 2 | 7 | 3 | 5 | 2 |

.010

**fig02**

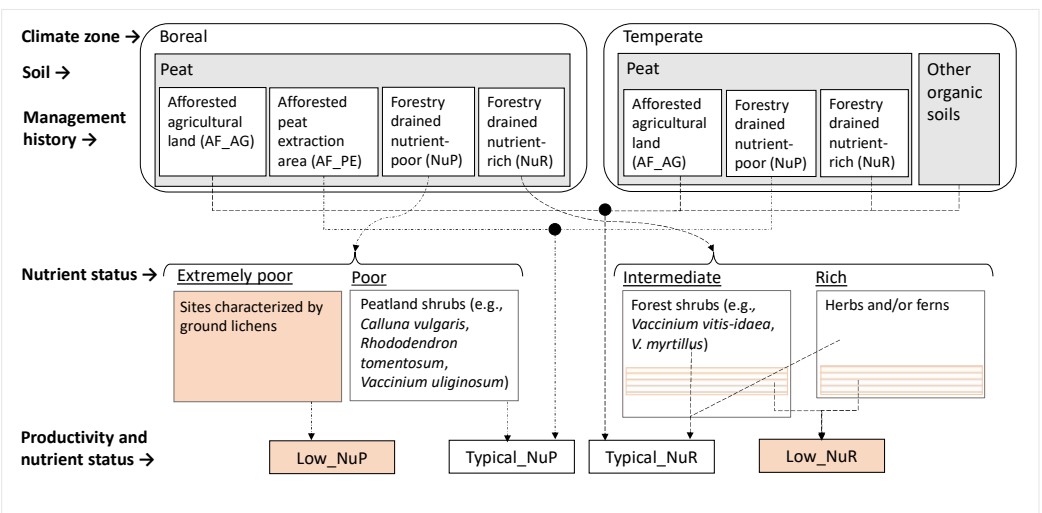

.015



**fig03**

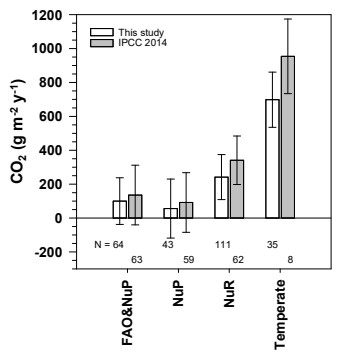 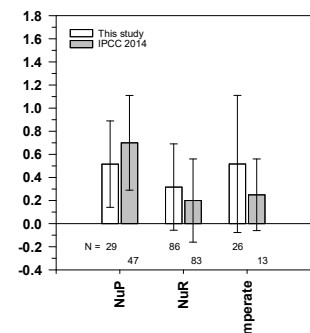 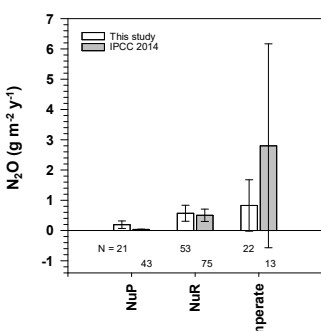

.020

**fig04**

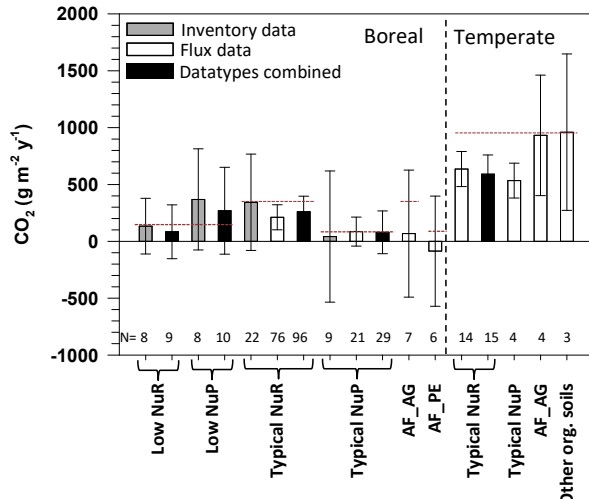

.025



**fig05**

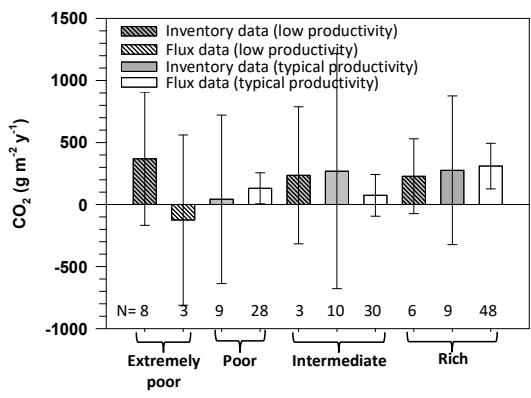
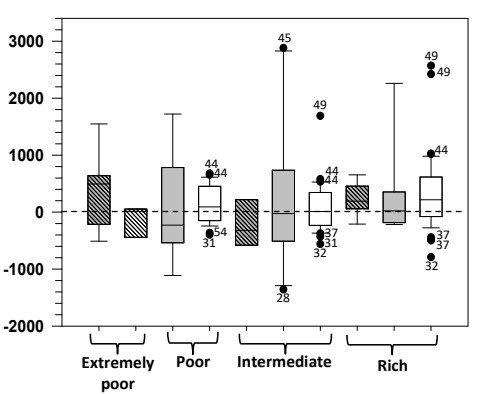

.030    **fig06**

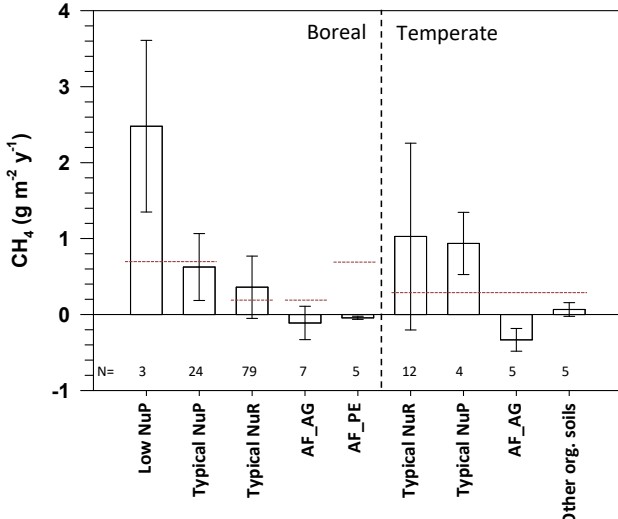





.035    **fig07**

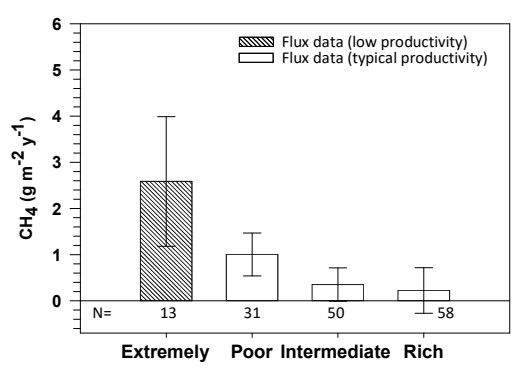
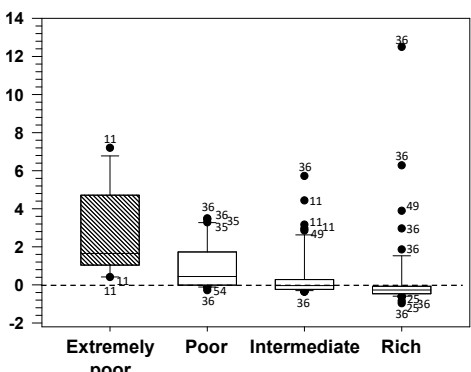

**fig08**

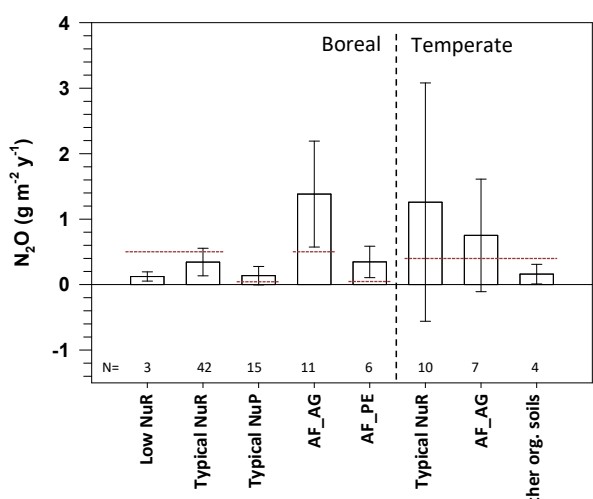

.040





**fig09**

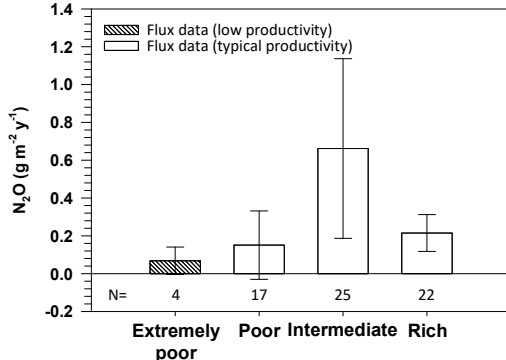 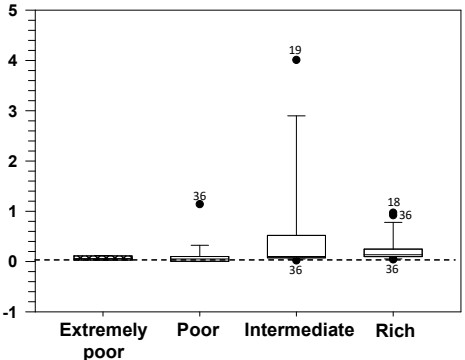

.045