# Peer review of "Reviews and syntheses: Greenhouse gas emissions from drained organic forest soils – synthesizing data for site-specific emission factors for boreal and cool temperate regions"

_Biogeosciences, 2023_

## Referee Comment (RC2)

A very sound, thorough and thus convincing study, which will contribute significantly to the precision of national GHG reporting in the case of disturbed peatlands, which are particularly relevant in this respect. There are very few aspects where something could be readjusted.

The categorization of monitoring sites was partly pragmatic, i.e. an important criterion was the availability of meaningful measurement, auxiliary and site data. This is perfectly fine in terms of the concern of the study. However, in order to use the limited capacity for measuring GHG fluxes efficiently in filling particularly relevant knowledge and data gaps, it would be helpful to conclude with a small summary overview of the site categories where particular action is needed. At present, this is done only sporadically and scattered (e.g., 575-577) . And in the case of the temperate zone, which is probably particularly affected (line 664), detailed information on the categories that require particularly urgent investigation is completely missing.

Several times in the manuscript, the methodological problems in GHG flux monitoring are mentioned ( e.g. lines 138-140, 633-638). However, this is done in a relatively abstract, stand-alone way, but not in concrete or specific connection with the discussion of the own results. Of course, this would not make sense everywhere either; but e.g. in the case of the lack of an explanation for high $N_2O$ emissions in mixed stands (lines 496-497), and especially in the context of the contradictory findings discussed in 4.3, methodological deficits could play a role. The importance of measurement aspects could also be briefly referred to in the conclusions with the suggestion that efforts to standardize, verify, and improve GHG flux monitoring (e.g., increased use of automated chamber approaches and isotope methods for flux separation) must also be continued.

Minor things that need clarification

Lines 58-60: Doesn't this also apply to Russia, Canada and Belarus?

Line 118: Complete/precise statement in parentheses, as not understandable at the moment.

Lines 496-498: Is there a connection in content between the two sentences? At the moment this is not apparent. Please specify.

---

## Author Comment (AC1)

**Referee1#**

**General remarks**

R1#: As described in the title, the authors made an excellent review and analysis of the drained organic forest soil site-specific EFs from boreal and cool temperate regions, aiming to review current Tier 1 default emission factors (EFs) provided by the IPCC (2014) Wetlands Supplement.

Even if the results were only modest and did not change much compared to the Tier 1 default EFs, the authors are to be complimented for this extensive work as it is a timely and very informative research in the context and preparation of next IPCC reports and UNFCCC NGHGIs revisions. I personally find this synthesis of great added value, not only for the scientific modelling community but also for the national inventory representatives. Such studies are highly needed in the context of reconciliation between reported and observed GHG estimates, and this is one great example of work combining the IPCC and scientific methodologies, and hopefully more such reviews will follow, not only for the LULUCF sector.

These results could be used to complement the Tier 1 reported estimates with higher Tiers 2 and 3 used by BU and TD modelling methodologies, to improve and provide a more accurate picture on actual emissions.

AR: We thank referee 1# for supportive general comments. we also agree that the results did not greatly differ from the findings of IPCC reporting (can be interpreted as "were modest"), which is actually favourable outcome in this context. The limitations observed in our study can be attributed to several factors. These include the variations in research methodologies, site-specific characteristics, and data availability across the diverse range of studies we analyzed. However, it is important to note that these limitations can serve as valuable lessons for enhancing future data collection and reporting practices.

**Specific remarks**

R1#: I would also add in the abstract and/or introduction information about the management status of the selected study-sites: are all managed sites, according the IPCC definitions (it is mentioned in the 2.1).
AR: While the term 'drained' organic forest soils already hints at the management status, we acknowledge the importance of clearly emphasizing the data's management status in the abstract and the introduction. We will make sure to incorporate this essential information.

R1#: Line 57: The authors wrote "Wetlands characterized by organic soils (we hereafter use 'wetlands')". I only found "wetlands" mentioned in the first two paragraphs of the Introduction. I personally think this is not needed, if else, please explain what was intended
AR: Good observation. The added definition had lost original purpose during former edits and thus can be omitted. Will be omitted.

R1#: Line 59: I saw in the table of figure 1 that few sites from the UK are also present, please add UK to the list of countries. I am somehow surprised that UK does not contribute more to this study.
AR: We have omitted listing of individual countries and instead provide main climate zone regions. Assessments of wetland areas in individual countries can be found in the provided references.

R1#: Line 71: the authors list the processes, I would add that all these lead to enhanced (GHG) emissions
AR: Depending of the process, changes can be enhancing, decreasing or even neutral. We have modified the text to point this out. 'Drainage and land-cover changes together alter rates in several processes: biomass growth, dead organic matter (litter) inputs into the soil, and litter and soil organic matter decomposition, leading to changes in GHG fluxes.

R1#: Line 78: did the authors check as well the 2019 IPCC Refinement (Chapter 7, Wetlands) for updated EF methodologies ?
AR: Thank you for noting this. We'll add this reference.

**R1#:** Line 160: do you only refer to "b) comparability of CO2 EFs" ? how about CH4 and N2O EFs?

**AR:** In this point-"b" it is not possible to do comparisons to CH4 and N2O due to inventory method characteristics (peat mass based C-differences conversion into CO2equivalents can be performed, but not for other GHGs). We will increased clarity in the text by adding name of gases compared in the point-'a' where all three major GHGs are compared between IPCC categories and categories developed in this study.

**R1#:** 2.1 Criteria for data selection paragraph: it is appreciated that authors follow the site section as defined by the IPCC

**AR:** Thank you.

**R1#:** Title 2.4: perhaps authors meant "different"

**AR:** Will be revised as suggested.

**R1#:** Line 369: I would delete the last words "in this study"

**AR:** We wish to keep this small detail in order to specify study where this observation was made. Well add another word to emphasize this.

**R1#:** Line 373: except for a removal

**AR:** Will be revised as suggested.

**R1#:** Line 377: except for the typical ..

**AR:** Will be revised as suggested.

**R1#:** Title 3.3: pretty long, I would add a comma after weather and delete "and"

**AR:** Will be revised as suggested.

**R1#:** Line 447: How about correlations with WT and soil temperature? I saw the WT is discussed in paragraph 4.1 but I would add it here as well

**AR:** Unfortunately, this comparison could not be performed due to inconsistent (/graphic form, /missing) WT reporting in publications.

**R1#:** Line 476: "This may be explained by the WT that, unfortunately, was not so generally available in the publications that it could have been used in the models" I would mention about this in the 4.4 data issues

**AR:** Good point. We will add WT into the list of inconsistencies in reporting.

**R1#:** Line 498: I totally agree that more research is needed regarding management sites. In the IPCC methodologies, only managed land is reported, therefore almost the entire EU is considered managed.

**AR:** We agree.

**R1#:** Title 4.2: perhaps authors meant "different"

**AR:** Will be revised as suggested.

**R1#:** Line 540: do authors know if this land was previously fertilized and for how long? Since how many years is afforested ??

**AR:** Status of fertilization was not necessarily reported but it is likely that all agricultural lands have been fertilized and peat extraction have not been fertilized prior to LUC. It would be interesting topic for assessing how GHG emissions develop during first decades after afforestation, but that would be topic to other assessment. We followed IPCC guidance for data selection where least 20 years as forest land after LUC is the criteria (provided in section 2.1 Criteria for data selection).

**R1#:** Title 4.3: the title is ab it long, perhaps remove the means vs medians and explain it in few lines within the paragraph: "in this section we compare the means vs medians of CO2 estimates from...."

**AR:** We acknowledge that title is somewhat long, but still wish to explain the content in the title for easier finding of the content.

R1#: Lines 633-639: as previously mentioned, perhaps good to add here the scarcity of WT information (from line 475)

AR: We totally agree and add more details in the text.

R1#: Conclusions: as already stated in the general remarks, it is a very useful study with improved ranges of uncertainty

AR: Thank you.

R1#: Table 1: I would also add in brackets the extensively used abbreviations from the main text (e.g., NuP, NuR etc.)

AR: We'll add respective category names used in this study as footnotes in the table.

---

## Author Comment (AC2)

**Referee2#**

R2#:

A very sound, thorough and thus convincing study, which will contribute significantly to the precision of national GHG reporting in the case of disturbed peatlands, which are particularly relevant in this respect. There are very few aspects where something could be readjusted.

**AR:** We thank referee 2# for the encouraging general comments.

R2#:

The categorization of monitoring sites was partly pragmatic, i.e. an important criterion was the availability of meaningful measurement, auxiliary and site data. This is perfectly fine in terms of the concern of the study. However, in order to use the limited capacity for measuring GHG fluxes efficiently in filling particularly relevant knowledge and data gaps, it would be helpful to conclude with a small summary overview of the site categories where particular action is needed. At present, this is done only sporadically and scattered (e.g., 575-577) . And in the case of the temperate zone, which is probably particularly affected (line 664), detailed information on the categories that require particularly urgent investigation is completely missing.

**AR:** We'll add more detailed description of the knowledge and data gaps in the conclusions.

R2#:

Several times in the manuscript, the methodological problems in GHG flux monitoring are mentioned ( e.g. lines 138-140, 633-638). However, this is done in a relatively abstract, stand-alone way, but not in concrete or specific connection with the discussion of the own results. Of course, this would not make sense everywhere either; but e.g. in the case of the lack of an explanation for high N2O emissions in mixed stands (lines 496-497), and especially in the context of the contradictory findings discussed in 4.3, methodological deficits could play a role. The importance of measurement aspects could also be briefly referred to in the conclusions with the suggestion that efforts to standardize, verify, and improve GHG flux monitoring (e.g., increased use of automated chamber approaches and isotope methods for flux separation) must also be continued.

**AR:**

Thank you for pointing this out. We have analyzed the methodological problems in detail in an earlier paper (Jauhiainen et al. 2019), and thus did not include much of that discussion in this paper. However, while analyzing this comment we realized that we did not really tell this to the readers of this paper! We'll both state this at the end of the Introduction, and add short notes in the locations pointed out by the Reviewer. However, for some issues, such as the anomalies related to high N2O emissions from mixed stands, or high CO2 emission values from poor sites based on inventory methods but not flux methods, we have found no plausible explanations in the data. In such cases we just acknowledge that. We hope that these revisions can be found satisfactory, but will be happy to do further revision if deemed useful.

Minor things that need clarification

Lines 58-60: Doesn't this also apply to Russia, Canada and Belarus?

Line 118: Complete/precise statement in parentheses, as not understandable at the moment.

Lines 496-498: Is there a connection in content between the two sentences? At the moment this is not apparent. Please specify.

**AR:**

- Lines 58-60: We'll omit listing of individual countries and instead provide main climate zone regions. Assessments of wetland areas in individual countries can be found in the provided references.
- Line 118: We agree that this added detail makes the sentence too complicated and decided to omit it.
- Lines 496-498: We agree that this is complicated text structure. We'll modify the text for creating the needed connection in two sentences and a better text flow. Two supporting references will be added.

---

## Author Response (AR2)

Dear Sirs,

We would like to express our gratitude for bringing to our attention the locations in need of technical corrections. The suggested changes have been implemented as requested for the first three items, and an alternative modification has been made for the fourth request.

L57: Delete double "in" -> "are found in boreal…"
AR: Done as requested.

L117: Insert space between NO3- and under.
AR: Done as requested.

L118: Change "produced from NO3– denitrification back to NH4+ in such reductive conditions" to "produced during the dissimilatory reduction of NO3- to NH4+ (DNRA)", since this process is not called denitrification.
AR: Done as requested.

L501-502: Change "and currently, there is data from a very limited number of sites concerning, e.g., impacts of different cuttings" to "and currently there are only data from a very limited number of sites, e.g., on the effects of different thinning measures"
AR: Since conventional forest logging options include thinning and clear-felling (as applied in the studies in the provided two references), the wording has been slightly modified to include clear-felling as well. The revised phrase is: "and currently there are only data from a very limited number of sites, e.g., on the effects of different tree harvesting intensities"

In addition, we have removed some double spaces that were noticed and added two previously missing links to the references.

We hope that these modifications to the text are useful and align with your requests.

Sincerely,
Jyrki Jauhiainen
On behalf of the author team